# Methamphetamine Enhancement of HIV-1 gp120-Mediated NLRP3 Inflammasome Activation and Resultant Proinflammatory Responses in Rat Microglial Cultures

**DOI:** 10.3390/ijms25073588

**Published:** 2024-03-22

**Authors:** Debashis Dutta, Jianuo Liu, Enquan Xu, Huangui Xiong

**Affiliations:** Department of Pharmacology and Experimental Neuroscience, University of Nebraska Medical Center, Omaha, NE 68198, USA; jialiu@unmc.edu (J.L.); xu.e@wustl.edu (E.X.)

**Keywords:** methamphetamine, HIV-1gp120, NLRP3 inflammasome, microglia, neuroinflammation

## Abstract

Human Immunodeficiency Virus type 1 (HIV-1)-associated neurocognitive disorders (HANDs) remain prevalent in HIV-1-infected individuals despite the evident success of combined antiretroviral therapy (cART). The mechanisms underlying HAND prevalence in the cART era remain perplexing. Ample evidence indicates that HIV-1 envelope glycoprotein protein 120 (gp120), a potent neurotoxin, plays a pivotal role in HAND pathogenesis. Methamphetamine (Meth) abuse exacerbates HANDs, but how this occurs is not fully understood. We hypothesize that Meth exacerbates HANDs by enhancing gp120-mediated neuroinflammation. To test this hypothesis, we studied the effect of Meth on gp120-induced microglial activation and the resultant production of proinflammatory cytokines in primary rat microglial cultures. Our results show that Meth enhanced gp120-induced microglial activation, as revealed by immunostaining and Iba-1 expression, and potentiated gp120-mediated NLRP3 expression and IL-1β processing and release, as assayed by immunoblotting and ELISA. Meth also augmented the co-localization of NLRP3 and caspase-1, increased the numbers of NLRP3 puncta and ROS production, increased the levels of iNOS expression and NO production, and increased the levels of cleaved gasderminD (GSDMD-N; an executor of pyroptosis) in gp120-primed microglia. The Meth-associated effects were attenuated or blocked by MCC950, an NLRP3 inhibitor, or Mito-TEMPO, a mitochondrial superoxide scavenger. These results suggest that Meth enhances gp120-associated microglial NLRP3 activation and the resultant proinflammatory responses via mitochondria-dependent signaling.

## 1. Introduction

The global epidemic of Human Immunodeficiency Virus type 1 (HIV-1) infection and acquired immunodeficiency syndrome (AIDS) remain worldwide public health issues. After more than 40 years of intensive research, we are still far from an HIV-1 cure due to viral reservoirs [1,2], and there is no successful vaccine available against HIV-1 despite enormous attempts [3,4,5]. Although the introduction and widespread availability of combined antiretroviral therapy (cART) have converted HIV-1 disease from a death sentence into a manageable chronic illness [6,7,8,9,10], HIV-1-associated neurocognitive disorders (HANDs) remain prevalent in infected individuals. The mechanisms underlying HAND prevalence in the cART era are not fully understood. A plethora of evidence indicates that these mechanisms are dependent on multiple factors, including but not limited to virus persistence in the brain, reduced CNS penetration of cART, viral protein neurotoxicity, chronic neuroinflammation, and increased life expectancy [11,12,13,14], as well as comorbid factors such as drugs of abuse. Amongst these disease-inciting factors, viral proteins and drugs of abuse play essential roles in HAND pathogenesis and prevalence.

HIV-1 envelope glycoprotein protein 120 (gp120) is a potent neurotoxin that plays a pivotal role in HAND pathogenesis. Shed off from virions and released from infected cells [13,15], gp120 accumulates in the cerebrospinal fluid and brain tissue in a significant amount and causes neuronal damage both in vitro and in vivo [16,17]. Studies have shown that gp120, on the one hand, induces neuronal apoptosis and synaptic and dendrite dysfunction [18,19] and, on the other hand, triggers immune activation and the resultant production of neurotoxic molecules, as well as inflammasome-dependent pyroptosis [20], leading to the development of HANDs [15,21,22]. The neurocognitive impairment observed in patients with HANDs can be attributed to the direct and indirect neurotoxic effects of gp120 and are frequently associated with and worsened by abuse of recreational drugs such as methamphetamine (Meth) [23,24].

Meth is a potent and highly addictive psychostimulant that is frequently used by HIV-1-infected individuals [25]. Meth abuse not only increases the risk of HIV-1 transmission but also augments HIV-1-associated neurocognitive impairments [26,27,28]. Ample evidence indicates that Meth exacerbates HANDs [23,29,30]. While many studies have focused on the individual effects of Meth on the CNS, far fewer have investigated its comorbid influence on HAND pathogenesis. We have previously reported that Meth augmented the gp120-induced enhancement of microglial outward K^+^ current, leading to increased production of proinflammatory molecules and consequent neuronal injury [31]. We have also shown, in another study, that the microglial nucleotide-binding domain, leucine-rich-containing family, pyrin domain-containing-3 (NLRP3) inflammasome was involved in Meth-induced enhancement of gp120 inhibition of long-term potentiation, a widely accepted synaptic mechanism for learning and memory, implying a potential mechanism for the Meth-induced exacerbation of HANDs seen clinically [32]. 

Inflammasomes are cytosolic multiprotein signaling complexes that trigger the activation of inflammatory caspases and the maturation of IL-1β. They are critical for the host’s immune defense against microbial infection and cell injury. Among various inflammasome complexes, NLRP3 is an extensively studied and well-characterized inflammasome [33,34] which recognizes multiple stimuli via NOD-like receptor (NLR) and serves as a platform for caspase-1 activation. Activation of the NLRP3 inflammasome requires two signals: The first one (priming signal) activates the transcription factor NF-κB, leading to the upregulation of NLRP3 and pro-IL-1β. The second one (activation signal) consists of various stimuli that promote the assembly of ASC and procaspase-1 and result in the activation of the NLRP3 inflammasome and caspase-1 [35,36,37,38]. The activated caspase-1 cleaves pro-IL-1β, pro-IL-18, and gasderminD (GSDMD), resulting in the release of matured IL-1β, IL-18, and GSDMD-N (a N-terminal fragment of GSDMD) and consequent inflammation and pyroptosis [34,36,39]. It is our hypothesis that Meth exacerbates HANDs via the augmentation of HIV-1-associated microglial inflammasome activation and resultant proinflammatory responses. To test this hypothesis, we investigated the effects of Meth on NLRP3 inflammasome activation and consequent proinflammatory cytokine production in gp120-primed rat microglial cultures. Our results show that Meth enhances gp120-stimulated microglial activation and the resultant cytokine production via mitochondria-dependent NLRP3 inflammasome activation. 

## 2. Results

### 2.1. Meth-Induced Enhancement of HIV-1 gp120-Induced IL-1β Processing and Release

IL-1β processing and release are stringently regulated by inflammasomes [40]. Inflammasome activation occurs in two steps: priming (first signal: transcription and expression of inflammasome components) and processing (second signal: assembly of inflammasome components) [40,41]. Although HIV-1 gp120 is known to signal both steps [20], an additional second signal can further enhance the activation of gp120-primed inflammasomes. We have previously demonstrated that treatment of microglia with Meth alone had no significant effects on the IL-1β transcript, suggesting that Meth does not work as the first signal for inflammasome activation [42]. Consistently with our previous results, the treatment of microglia with Meth alone at different concentrations (6, 18, and 50 μM) failed to produce significant effects on IL-1β processing as detected by immunoblotting (Figure 1A,B) and IL-1β production as measured by ELISA (Figure 1C). To test the effects of gp120 on inflammasome activation, microglial cells were treated with gp120 at various concentrations (250, 500, and 1000 pM), and concentration-dependent responses were observed. As evident from the Western blot results, significant IL-1β processing occurred at 500 pM (2.7-fold, *p* < 0.01) and 1000 pM (3-fold, *p* < 0.01) compared with untreated controls (Figure 1D,E). These results were parallel to the results of IL-1β release in the culture supernatant as measured by ELISA (Figure 1F), illustrating concentration-dependent IL-1β release mediated by gp120. As 500 pM gp120 increased IL-1β processing and release and 50 μM Meth had no significant effect, we examined if Meth could enhance gp120 effects on IL-1β processing and release when applied in combination. The results showed that Meth (50 µM) enhanced IL-1β processing (2.6-fold, *p* < 0.01) and release (3.7-fold, *p* < 0.001) in gp120-primed microglia compared with the results from solely gp120-primed microglia (Figure 1G–I). These results provide us with the optimum concentrations of Meth (50 µM) and gp120 (500 pM) for testing the Meth-induced enhancement of gp120 effects on microglial inflammasome activation in this study. 

As evident from the Western blot results, significant IL-1β processing occurred at 500 pM (2.7-fold, *p* < 0.01) and 1000 pM (3-fold, *p* < 0.01) compared with the untreated control (Figure 1D,E). These results were further supported by IL-1β release in the culture supernatant measured by ELISA (Figure 1F). Significant dose–response of IL-1β release was observed at 500 pM (3.4-fold, *p* < 0.01) and at 1000 pM (5.2-fold, *p* < 0.001) compared with the untreated control sample. As 500 pM gp120 was the “lowest” concentration to produce significant effects on IL-1β processing and release, this concentration was adapted in the experiments exploring Meth-induced enhancement of gp120-mediated microglial proinflammatory responses. 

### 2.2. Meth Augments gp120-Primed Microglia Activation 

Ionized calcium-binding adaptor molecule 1 (Iba-1) is a microglia-specific marker, and its expression is known to increase upon microglial activation [43,44]. Immunofluorescence microscopy and immunoblotting revealed that unstimulated rat microglia expressed a low level of Iba-1. In gp120-treated microglia, a remarkable increase in Iba-1 staining was observed (Figure 2A). The latter was further enhanced after the treatment of gp120-primed microglial cells with Meth (Figure 2A). In agreement with immunofluorescence staining, the Western blot results of cell lysate show an increased level of Iba-1 in microglia treated with gp120, which was increased by Meth (Figure 2B,C). Statistical analyses revealed that the levels of Iba-1 expression were significantly (*p* < 0.01) increased when microglial cells were treated with gp120 and Meth in combination compared with those treated with gp120 alone or the untreated control. These results demonstrate Meth-induced enhancement of gp120-induced microglial activation.

### 2.3. Effect of Meth on NLRP3 Co-Localization and Puncta Formation

The NLRP3 inflammasome plays an important role in microglial activation [34]. To evaluate the effect of Meth on NLRP3 inflammasome activation, we examined the co-localization of NLRP3 and caspase-1 in gp120-primed microglia. Our results show increased NLRP3/caspase-1 co-localization upon Meth treatment (Figure 3), indicating Meth-induced enhancement of microglial NLRP3 inflammasome activation, which was further validated by NLRP3 puncta formation, as visualized by immunofluorescence. We observed a significant increase in NLRP3 puncta formation upon Meth treatment in gp120-primed microglia (Figure 4A,B). The pretreatment of microglial cultures with MCC950 (a selective inhibitor of NLRP3 inflammasome) or Mito-TEMPO (a mitochondria ROS inhibitor) attenuated Meth-induced increase in NLRP3 puncta formation (Figure 4B), indicating NLRP3 activation of and involvement in the induction of ROS upstream signaling. Additionally, Meth was also found to augment the gp120-primed increase in NLRP3 expression (Figure 4C,D). The pretreatment of microglial cultures with MCC950 significantly blocked the Meth/gp120-associated increase in NLRP3 expression. However, partial blockade of the latter was observed when microglial cultures were pretreated with Mito-TEMPO. 

### 2.4. Meth-Induced Enhancement of Proinflammatory Cytokine Production in gp120-Primed Microglia 

It is well established that the upregulation of proinflammatory cytokines plays multiple roles in both neurodegeneration and neuroprotection. To examine if the Meth-induced enhancement of gp120-primed microglial NLRP3 inflammasome activation could increase cytokine production, we measured the levels of TNF-α, IL-1β, IL-6, and IL-18 in the culture supernatants by ELISA. Significant increases in IL-1β (~4.35 fold), TNF-α (~10 fold), IL-6 (~7.81 fold), and IL-18 (~55 fold) in gp120-primed and Meth-treated microglia (Figure 5A–D) compared with the supernatants collected from gp120-primed microglia were observed, suggesting that the Meth-induced enhancement of proinflammatory cytokine release is responsible for neuronal injury [45]. In addition, Meth was found to enhance mitochondrial total ROS production (Figure 6A,B). The Meth-associated increase in cytokine production was blocked by pretreatment of microglial cells with MCC950 or Mito-TEMPO, suggesting the involvement of the NLRP3 inflammasome and its upstream mitochondrial ROS signaling in Meth-associated enhancement of cytokine release. 

### 2.5. Meth-Induced Increase in iNOS Expression and NO Production

Studies have shown that activation of microglia causes overproduction of nitric oxide (NO) by inducible nitric oxide synthase (iNOS), resulting in neuroinflammatory processes [46,47,48]. To assay the levels of NO production and iNOS expression in gp120-primed microglia treated with Meth, we measured NO production in microglial culture supernatants by ELISA and iNOS expression in microglial lysate by Western blot. The results show that the application of Meth to gp120-primed microglial cultures significantly increased NO production (Figure 7A) and iNOS expression (Figure 7B,C). 

### 2.6. Meth Increased GSDMD-N Production in Microglial Cells Primed with gp120

Pyroptosis is a form of proinflammatory programmed cell death mediated by caspase-1-cleaved pore-forming protein gasdermin D (GSDMD) [49]. The N-terminal proteolytic fragment of GSDMD (GSDMD-N) is an executor of pyroptosis and is required for IL-1β release [50]. To explore the involvement of NLRP3-dependent pyroptosis in Meth-induced enhancement of gp120-associated pathophysiology, we examined the expression levels of GSDMD-N in microglial cultures. The treatment of microglial cells with gp120 led to cleavage of GSDMD and increased production of GSDMD-N (Figure 8A). The Meth-induced increase in GSDMD-N production in microglial cells primed with gp120 implies the occurrence of inflammatory pyroptosis triggered by Meth. The addition of MCC9500 or Mito-TEMPO to microglial cultures attenuated the Meth-/gp120-associated increase in GSDMD-N production, implying activation of NLRP3/caspase-1 by Meth/gp120.

## 3. Discussion

Although cART has significantly decreased the spectrum of disease morbidities, including profound dementia, more subtle forms of HIV-1-associated neurocognitive disorders (HANDs) remain prevalent [14,51,52]. Virus persists in the brain at low levels, often in a latent or restricted manner. Immune activation and neuroinflammation, which are linked to viral proteins and drugs of abuse, continue to play pivotal roles in HAND pathogenesis. Meth abuse exacerbates the HANDs seen clinically, and the mechanism(s) underlying this remains unclear [23,53,54]. To understand how Meth exacerbates HANDs, we studied the augmentation effects of Meth on HIV-1 gp120-induced microglial NLRP3 inflammasome activation and the resultant proinflammatory cytokine production. Our results reveal that these effects were exerted via mitochondria-dependent NLRP3 inflammasome activation. 

The NLRP3 inflammasome is a critical component of the innate immune system that mediates caspase-1 activation and proinflammatory cytokine production in response to diverse stimuli and multiple biomolecules, including but not limited to viral proteins and drugs of abuse. Typically, two independent signals are required to fully activate the NLRP3 inflammasome [35,42]. To understand whether Meth augments gp120-induced NLRP3 inflammasome activation in microglia, we previously examined the effects of Meth on lipopolysaccharide (LPS; a known priming signal)-induced NLRP3 inflammasome activation in rat microglial cultures. We observed that Meth could augment the pre-existing inflammatory stimulation and produce an increase in IL-1β maturation and release in an NLRP3 inflammasome-dependent manner. In the present study, we substituted LPS with HIV-1 gp120 to reflect disease conditions in human subjects and investigated the enhancement effects of Meth on HIV-1 gp120-induced rat microglial activation and the resultant inflammatory responses, focusing on the involvement of the NLRP3 inflammasome. We observed that Meth enhanced NLRP3 inflammasome activation and proinflammatory cytokine production in HIV-1 gp120-primed rat microglial cells. The enhancement effects were attenuated or blocked by the addition of the NLRP3 inflammasome inhibitor MCC950 and/or the mitochondria-targeted antioxidant Mito-TEMPO, suggesting the involvement of the mitochondria-dependent NLRP3 inflammasome in the Meth-induced enhancement of gp120-stimulated microglial activation and proinflammatory cytokine production. 

Meth is one of the most abused drugs among individuals infected with HIV-1 [25]. Ample evidence indicates that Meth abuse exacerbates cognitive deficits and neurodegenerative abnormalities in HIV-1-infected patients and animal models [27,28,55,56]. To explore the impact of Meth on HIV-1 gp120-induced microglial NLRP3 activation, we first examined the individual effects of Meth and gp120 on NLRP3 inflammasome activation in primary rat microglial cultures. Consistently with our previous observations [42], Meth had no significant impact on pro-IL-1β/IL-1β expression and IL-1β release at three different doses (6, 18, and 50 µM), indicating that this compound may not work as the first signal for inflammasome activation [42]. However, HIV-1 gp120 significantly increased pro-IL-1β/IL-1β expression and IL-1β release at concentrations of 0.5 nM and higher, suggesting that gp120 could cause NLRP3 inflammasome activation. When tested in combination, 50 µM Meth was found to enhance the gp120-mediated increase in pro-IL-1β/IL-1β expression and IL-1β release, implying Meth-induced enhancement of gp120-associated NLRP3 inflammasome activation.

HIV-1 gp120 plays a vital role in the HAND pathogenesis. It causes immune activation and the resultant production of proinflammatory cytokines and inflammasome-dependent pyroptosis, in addition to its direct toxic effects on neural cells. The neurotoxic effects of HIV-1gp120 could be augmented by drugs of abuse, such as Meth. The mechanisms underlying the Meth-induced enhancement of HIV-1gp120-associated neurotoxicity are multifaceted and include but are not limited to the activation of the microglial NLRP3 inflammasome. Studies have shown that the NLRP3 inflammasome is involved in HIV-1 gp120-associated microglial activation [20,57] and the resultant neuronal injury [32]. In agreement with the abovementioned studies, our results show that Meth enhanced pro-IL-1β/IL-1β expression, processing, and release in HIV-1 gp120-primed microglial cultures which were attenuated or blocked by MCC950, a specific NLRP3 inflammasome inhibitor, demonstrating Meth-induced enhancement of gp120-primed microglial activation via NLRP3 inflammasome signaling in cultured rat microglial cells.

The treatment of microglia with Meth significantly increased the production levels of IL-1β, TNF-α, IL-6, and IL-18 in gp120-primed microglia compared with those subjected to individual treatments. The production of these cytokines was significantly reduced by the pretreatment of microglia with Mito-TEMPO and MCC950, suggesting an involvement of mitochondria and NLRP3 inflammasome in the Meth-associated increase in cytokine production. As mitochondria represent an important source of ROS, we measured total ROS activation and observed a significant increase after the treatment of microglia with gp120 and Meth, indicating that ROS may play a role in inflammasome activation signaling [58]. In addition to ROS activation, Meth-associated increase in inducible nitric oxide synthase (iNOS) expression and increase in nitric oxide (NO) production were detected in gp120-primed microglial lysate and culture supernatants, respectively. The increased production of NO could be one of the mechanisms underlying the Meth-induced exacerbation of HANDs seen clinically. 

The Meth-induced enhancement of HIV-1gp120-primed NLRP3 inflammasome activation was supported by experimental results demonstrating the co-localization of NLRP3 inflammasome with its downstream effector protein, caspase-1, and the formation of NLRP3 puncta as visualized by immunofluorescence microscopy. We observed that the co-localization of NLRP3 and caspase-1 was enhanced upon Meth treatment of the gp120-primed microglial cells, a sign of Meth-induced enhancement of NLRP3/caspase-1 activation. As NLRP3 activation leads to the assembly of NLRP3, ASC, and caspase-1 and the formation of the NLRP3 inflammasome complex, a micron-sized dense structure known as puncta, Meth was found to increase the numbers of NLRP3 puncta-positive cells, a hallmark of inflammasome activation [59]. The treatment of microglial cells with MCC950 significantly decreased NLRP3/caspase-1 co-localization and puncta formation, indicating an involvement of the NLRP3 inflammasome in the Meth-associated enhancement of NLRP3/caspase-1 colocalization and puncta formation. In addition to NLRP3 puncta quantification, we measured the levels of NLRP3 expression using immunoblotting and found a significant increase in NLRP3 expression in gp120/Meth-treated microglial cells, which was significantly restored upon the pretreatment of the cells with MCC950. The detection of increased colocalization and puncta formation of the NLRP3 inflammasome complex inside a cell after the Meth treatment of HIV-1gp120-primed microglial cells strongly supports the Meth-induced enhancement of HIV-1gp120-associated microglial activation.

Canonically, the inflammasome can be activated in response to various upstream signals. As the primary mediator for pro-IL-1β maturation, NLRP3 inflammasome activation is accompanied by the processing of pro-IL-1β, the cleavage of caspase-1, and ASC protein aggregation. After sequential stimulation with HIV-1gp120 and Meth, NLRP3 and IL-1β were cleaved to their activated forms, as illustrated by their upregulated expression levels and increased proinflammatory cytokine production (e.g., IL-1β, IL-6, IL-18, and TNFα). The increase in proinflammatory cytokine production led to pyroptosis, as demonstrated by the increased expression of GSDMD-N, a central player in executing pyroptosis, the cell death pathway downstream of inflammasome activation [60]. These results are in agreement with the classical cellular pattern of inflammasome activation, in which principal components redistribute from dispersed to clustered arrangements and promote the restoration of caspase-1 enzyme activity after the cross-cleavage process [61]. 

Lastly, it is worth pointing out that all experiments were carried out on rat microglial cultures instead of human microglial cells due to not having been able to obtain sufficient human microglial cells for this study. Thus, the findings reported here may have a limitation in implying disease conditions in human subjects. However, Meth was found to exacerbate HANDs in HIV-1-infected individuals, and the results we observed, which indicate Meth-induced enhancement of gp120-associated microglial NLRP3 activation and resultant proinflammatory responses, may implicate the potential mechanisms underlying the Meth-induced exacerbation of HANDs seen clinically.

## 4. Materials and Methods

### 4.1. Materials

Full-length HIV-1 envelope glycoprotein 120 from a Clade B virus (HIV-1_MN_ gp120, Cat# IT-001-002MNp) was purchased from Immune Technology Corp. (New York, NY, USA) and stored at −80 °C in a freezer in 100 nM aliquots. Methamphetamine was purchased from Sigma-Aldrich (St. Louis, MO, USA; Cat# M-8750) with DEA license # RX0374974. Lipopolysaccharide (LPS; from *Escherichia coli* 0111:B4; Cat# L4391) was also procured from Sigma-Aldrich (Saint Luis, MO, USA). MCC950 (Cat# 538120) and Mito-TEMPO (Cat# AL-X-430-150-M005) were obtained from Enzo Life Sciences (Farmingdale, NY, USA). All other chemicals, unless otherwise specified, were purchased from Sigma-Aldrich.

### 4.2. Animals

For the isolation of microglia from neonates, pregnant female Sprague-Dawley (SD) rats were purchased from Charles River Laboratories (Wilmington, MA, USA). The animals were kept in the university animal house at constant temperature (22 °C) and relative humidity (50%) under a regular light–dark cycle (light was turned on at 7:00 a.m. and off at 5:00 p.m.) with adequate access to food and water round the clock. All the animal use procedures in the study were strictly reviewed by the Institutional Animal Care and Use Committee (IACUC) of University of Nebraska Medical Center (IACUC No. 19-085-07-FC).

### 4.3. Isolation and Culture of Microglial Cells

Microglial cells were isolated from the cerebral cortex of postnatal (0–1 day old) SD rats as described previously [62]. Briefly, rat cortical tissues were dissected in cold Hank’s Balanced Salt Solution (HBSS; Cat# 14025-076; Life Technologies, Grand Island, NY, USA) and digested with 0.25% trypsin (Cat# 25200-056) and 200 Kunitz units/mL DNase (Cat# D4263; Sigma-Aldrich, Saint Luis, MO, USA) at 37 °C for 30 min. The digested tissues were then suspended in cold HBSS and filtered through 100 µM and 40 µm pore cellular strainers (BD Bioscience, Durham, NC, USA). The isolated cells (25 × 10^6^) were plated into T75 cm^2^ flasks (Cat# 90076; TPP, Tecchno Plastic Products AG, Trasadingen, Switzerland) in high-glucose Dulbecco’s modified Eagle’s medium (DMEM; Cat# 11965-084; Life Technologies, Grand Island, NY, USA) containing 10% fetal bovine serum (FBS; Cat#, A5670701; ThermoFisher Scientific, Asheville, NC, USA), 1×glutaMAX (Cat# 35050; Life Technologies, Grand Island, NY, USA), 1% penicillin/streptomycin (Cat# 30-002-Cl; Mediatech, Inc., Manassas, VA, USA), and 300 ng/mL macrophage colony-stimulating factor (M-CSF) supplied by the Department of Pharmacology and Experimental Neuroscience, University of Nebraska Medical Center. After 8–10 days in culture, the flasks were gently shaken, and the detached cells were collected and seeded onto 6-well (2 × 10^6^/well) and 12-well (1 × 10^6^/well) or 96-well plates (0.5 × 10^6^/well) based on the experimental requirements with M-CSF-free DMEM. The suspensory glial cells were removed 1 h after seeding by changing the culture media. The resultant cultures were 98–100% microglia as determined by staining with anti-CD11b (Cat# ab8879; Abcam, Cambridge, MA, USA), a marker for microglia.

### 4.4. Western Blotting

After priming with gp120 for 48 h, the microglial cells were treated with Meth for another 24 h. The cells were then lysed using RIPA lysis buffer (Cat# 89901; ThermoFisher Scientific, Asheville, NC, USA) for analyzing pro-IL-1β processing. An amount of 30 µg of total proteins was separated by 4–20% gradient PAGE (Cat# 4561094; Bio-Rad, Hercules, CA, USA) and transferred to nitrocellulose polyvinylidene difluoride (PVDF) membranes (Cat# 1620177; Bio-Rad, Hercules, CA, USA). The membranes were blocked with 3% bovine serum albumin (BSA; Cat#) in tris-buffered saline (TBS) and incubated overnight at 4 °C with rabbit polyclonal antibody for IL-1β (Cat# EPR21086; Abcam, Cambridge, MA, USA) at 1:500 dilution, mouse polyclonal antibody for NLRP3 at 1:1000 dilution (Cat# AG-20B-0014-C100; AdipoGen, San Diego, CA, USA), rabbit Iba-1 antibody (Cat# 016-20001; Fujifilm, Richmond, VA, USA) at 1:1000 dilution, or anti-mouse β-actin monoclonal antibody (1:5000; Cat# A5441; Sigma-Aldrich, Saint Louis, MO, USA). The washing buffer was TBS with 0.2% Tween (TBS-T). For iNOS expression, rabbit polyclonal antibody for iNOS (Cat# ab15323; Abcam, Cambridge, MA, USA) at 1:500 dilution was used. For pyroptosis analysis, rabbit monoclonal anti-GSDMD (Cat# ab219800; Abcam, Cambridge, MA, USA) was used. The secondary antibody was horseradish peroxidase (HRP)-conjugated anti-rabbit (Cat# 31460) or anti-mouse (Cat# 31430; Invitrogen, Rockford, IL, USA). The labeled proteins were visualized by the Pierce-enhanced chemiluminescence (ECL) system (Cat# 30496; Thermo Fisher Scientific, Waltham, MA, USA).

### 4.5. Enzyme-Linked Immunosorbent Assay (ELISA) Analysis

Secretion of IL-1β and other cytokines in the culture supernatants was assayed by ELISA. Microglia were primed with gp120 for 48 h and then stimulated with Meth at different concentrations for another 24 h. The cells were washed three times with PBS before the addition of Meth. The microglia were washed three times between these two treatments. To measure IL-1β release, the supernatant of Meth-treated microglia was collected at 24 h. Cytokines in supernatants were detected using the ELISA kit (Cat# DY501 for IL-1β, DY510 for TNFα, DY521-05 for IL-18, and DY506 for IL-6; R&D systems, Inc. Minneapolis, MN, USA). The experiments were performed following the manufacturer’s instructions. Briefly, the plates were coated with capture antibody overnight at room temperature; then, the reagent dilution buffer was used as the blocking reagent. After overnight incubation, the plates were washed three times with wash 1× buffer. The capture antibody-coated 96-well plates were incubated with the collected supernatants for 2 h at room temperature, followed by 2 h application of the detection antibody. After each step, the plates were washed using 1× wash buffer. Finally, the Streptavidin-HRP working solution was incubated for 20 min before substrate solution was added to each well for 20 min. Afterwards, the reaction was stopped with a stop solution, a reading was performed by using a Bio-Rad microplate reader with filters at 450/560 nm, and the result was calculated using a 4-parametric curve.

### 4.6. Measurement of Nitric Oxide (NO) Production

The production of nitrite was measured by the Griess reagent system (Cat# TB229; Promega, Madison, WI, USA) according to the manufacturer’s instructions. After treatment with gp120 and Meth, 50 µL aliquots of culture supernatant were collected from each treatment group. Then, 50 µL of sulfanilamide solution and 50 µL of NED solution were supplemented with the collected supernatants for 10 min in each case. The absorbance of the final samples was measured with a Bio-Rad microplate reader with filters at 560 nm. 

### 4.7. Fluorescent Dye Loading and Imaging

Fluorescent probes against ROS production with H2DCFDA were loaded onto the microglia that had been treated with Meth for an additional 24 h after priming with gp120 for 48 h. 5-(and-6)-chloromethyl-2′,7′-dichlorodihydrofluorescein diacetate (CM-H2DCFDA) (Cat# vC6827, Life Technologies, Eugene, OR, USA) was deployed to examine the intracellular ROS production. CM H2DCFDA enters cells passively, and the acetate groups of the probe were cleaved by intracellular esterase, leading to better cellular retention. After oxidation by reactive oxygen intermediates generated in response to Meth, the non-fluorescent H2DCFDA was converted to the highly fluorescent 2′,7′-dichlorofluorescein (DCF). The CM-H2DCFDA (5 μM) working solution was freshly made with pre-warmed DMEM and incubated on treated microglia at 37 °C for 30 min. The cells were then fixed with ice-cold 4% paraformaldehyde in PBS for 10 min and counterstained with 4′,6-diamidino-2-phenylindole (DAPI). To quantify the results, the microglia were seeded onto a 96-well black plate at a density of 0.25 × 10^6^/well, and the treatment procedure mentioned above was then repeated. After loading with CM-H2DCFDA, the intensities of fluorescent signals were evaluated by the microplate reader.

### 4.8. Immunocytochemistry

Immunocytochemistry was performed to quantify NLRP3 puncta as a readout of inflammasome activation. Microglia were seeded on coverslips in a 24-well plate at a density of 0.5 × 10^6^ cells per well. After priming with gp120 (0.5 nM) for 48 h, Meth treatment was administered for 24 h. Iba-1 staining was conducted to monitor microglial activation. The cells were fixed with 4% paraformaldehyde (PFA) for 10 min at room temperature. Then, the cells were blocked and permeabilized in PBS with 10% goat serum and 0.1% Triton X-100 for 15 min. The primary antibodies used included mouse polyclonal antibody for NLRP3 (Cat# sc-518122; Santa Cruz, CA, USA) and mouse monoclonal Iba-1 (Cat# sc-32725; Santa Cruz, Dallas, TX, USA) at 1:100 dilution, rabbit polyclonal antibody for ASC (Cat# sc-22514-R; Santa Cruz, Dallas, TX, USA) at 1:200 dilution, and caspase-1 (Cat# sc-514; Santa Cruz, Dallas, TX, USA) at 1:500 dilution. The microglia were identified by mouse monoclonal antibody CD11b at 1:500 dilution (Cat# ab8879; Abcam, Cambridge, MA, USA). The secondary antibodies used here were goat anti-rabbit Alexa 488 (1:1000, Cat# A-11034) and Alexa594 (1:1000 Cat# A-11012), and goat anti-mouse Alexa 488 (1:1000, Cat# A-11001) and Alexa594 (1:1000 Cat# A-11032) from ThermoFisher Scientific (Waltham, MA, USA).

### 4.9. Data Analyses

All data are expressed as means ± SDs unless otherwise indicated. Statistical analyses were performed by one-way ANOVA followed by post hoc Tukey’s multiple comparisons test (GraphPad Prism, version 9.4.1). A minimum *p*-value of 0.05 was chosen as the significance level for all tests. The densities of target Western blot bands were quantified using NIH Image J software (Version: Java 1.8.0_172, 64-bit) and standardized to β-actin band density. The percentage of microglia with NLRP3 puncta was estimated by microscopic scoring as per the description in the previous article by Rodrigues et al. [63]. In the fluorescent staining for total ROS production, nine field images were taken with a fluorescent microscope, and all cellular fluorescent intensities were quantified by Image J. The intensities of all individual cells in each field were averaged and transformed to fold changes against a control group. All experiments were performed in technical triplicate and biological triplicate unless otherwise specified. 

## 5. Conclusions

The present study demonstrated Meth-induced enhancement of proinflammatory responses in gp120-primed rat microglial cultures via NLRP3 inflammasome activation. Meth enhanced gp120-associated NLRP3 expression and IL-1β processing and release; it also increased the co-localization of NLRP3 with caspase-1 and ROS production. In addition, Meth increased the levels of iNOS expression and NO production, as well as the level of cleaved GSDMD-N, an executor of pyroptosis, in gp120-primed microglia. The Meth-associated effects were attenuated by MCC950, a NLRP3 inhibitor, or Mito-TEMPO, a mitochondrial superoxide scavenger. These results evidence Meth-induced enhancement of proinflammatory responses in gp120-primed rat microglial cultures via NLRP3 activation and mitochondria-mediated signaling (Figure 9).

## Figures and Tables

**Figure 1 ijms-25-03588-f001:**
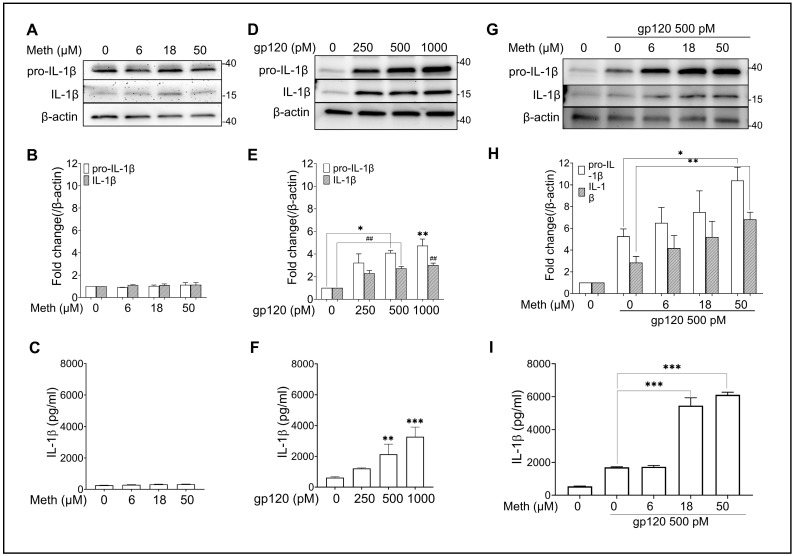
Meth-induced enhancement of HIV-1 gp120-induced IL-1β processing and release. The data illustrated in three columns were obtained from rat microglial cultures treated with Meth (left) and gp120 (central), with each compound being applied individually at different concentrations, or with one concentration of gp120 (500 pM) and different concentrations of Meth (right). The upper row displays the Western blot results of the expression levels of pro-IL-1β and IL-1β when microglia were treated with Meth (**A**) and gp120 (**D**) alone or 500 pM gp120 with Meth at different concentrations (**G**). Bar graphs in the middle row (**B**,**E**,**H**) show the densitometry quantitation of pro-IL-1β and IL-1β corresponding to the Western blots shown in the upper row. The lower row (**C**,**F**,**I**) shows the levels of IL-1β detected by ELISA from culture supernatants of microglia under different experimental conditions as indicated. ^##^ *p* < 0.01, * *p* < 0.05, ** *p* < 0.01, and *** *p* < 0.001 vs. untreated controls (**E**,**F**) or vs. gp120-treated controls (**H**,**I**). Data represent means ± SDs derived from three independent experiments in triplicate.

**Figure 2 ijms-25-03588-f002:**
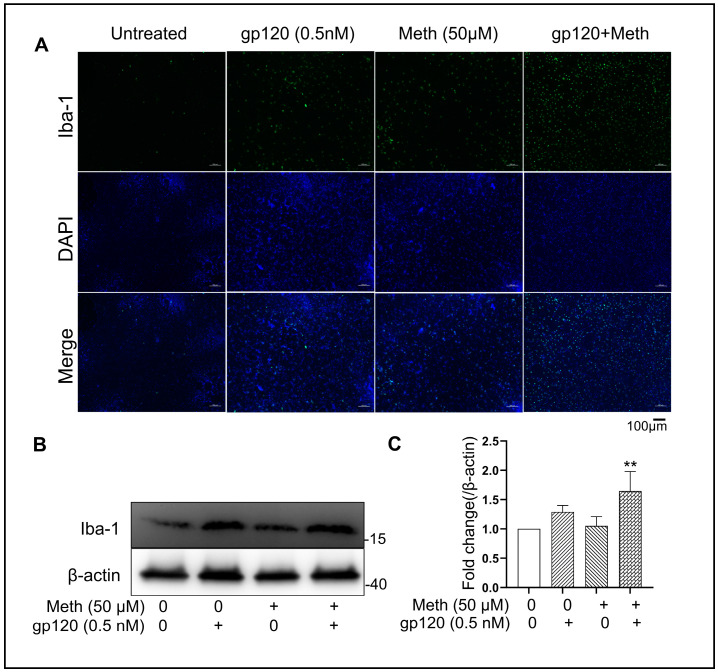
Meth augments gp120-primed microglial activation. (**A**) Immunofluorescence labeling of Iba-1 expression in microglia under different experimental conditions as indicated. Note that enhanced Iba-1 expression was detected in microglia treated with gp120 + Meth. (**B**) Western blot detection of Iba-1 protein expression and increased levels of Iba-1 expression were observed with gp120 + Meth treatment. (**C**) Image J densitometry analyses of Iba-1 protein expression in Western blots. Each bar represents average with standard deviation of three biological replicate experiments. ** *p* < 0.01: one-way ANOVA followed by Dunnett’s comparisons test. Scale bar equals to 100 µM.

**Figure 3 ijms-25-03588-f003:**
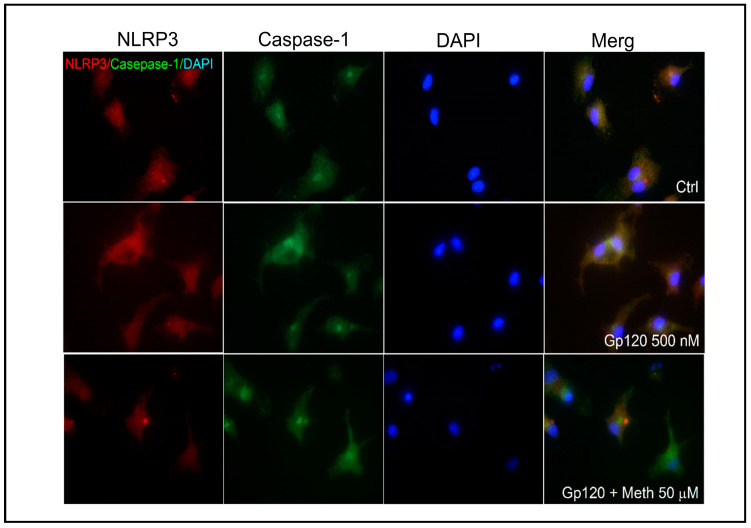
Meth-induced co-localization of NLRP3 and caspase-1 in gp120-primed microglia. Immunofluorescence images were taken from microglial cultures treated with gp120 (middle row) and gp120+ Meth (bottom row) after these were stained separately with primary antibodies against NLRP3 (red) and caspase-1 (green). Upper row refers to untreated controls. Note that Meth-induced co-localization of NLRP3 and caspase-1 is shown in merged images. Blue color indicates DAPI staining. Images were taken at 60× magnification.

**Figure 4 ijms-25-03588-f004:**
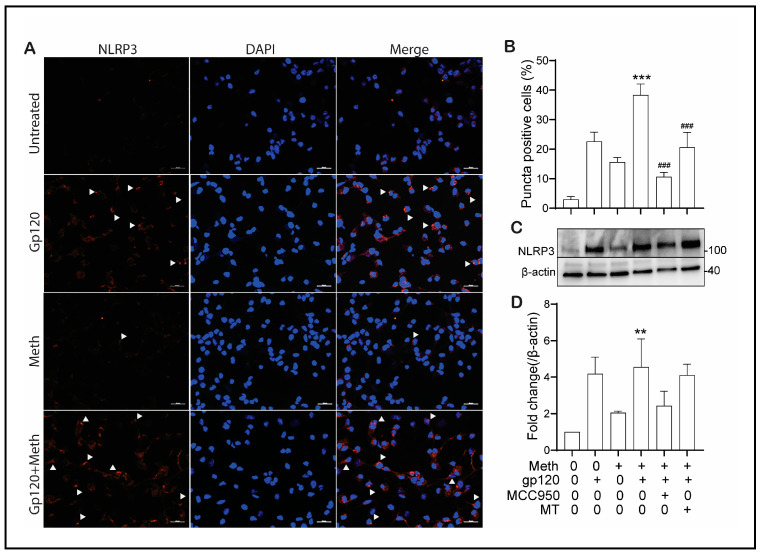
Meth triggers NLRP3 activation in gp120-primed microglia. Panel (**A**): Representative fluorescence microscopy images display NLRP3 puncta (red) labeled by anti-NLRP3 antibody (left column) under different experimental conditions as indicated. Cell nuclei were stained with DAPI (blue, middle column). Right column shows merged images. Panel (**B**): Microscopic scoring estimated the percentage of microglia containing NLRP3 puncta, shown as averages with standard deviations from three biological replicates, each performed with technical triplicates. Panels (**C**,**D**): Western blot detection of NLRP3 and corresponding densitometric analysis of NLRP3 under different treatment conditions, where densitometric values are averages with standard deviations from three independent biological replicate experiments, each experiment performed with technical triplicates. Note synergic effects of Meth and gp120 on increase in NLRP3 puncta-positive cells and NLRP3 expression levels and their significant blockade by MCC 950 (NLRP3 inflammasome activation blocker) or by Mito-TEMPO (MT; mitochondria-targeted antioxidant). *** *p* < 0.001, ^###^ *p* < 0.001, and ** *p* < 0.01: one-way ANOVA followed by Tukey’s multiple comparisons tests. Scale bar equals to 30 μM.

**Figure 5 ijms-25-03588-f005:**
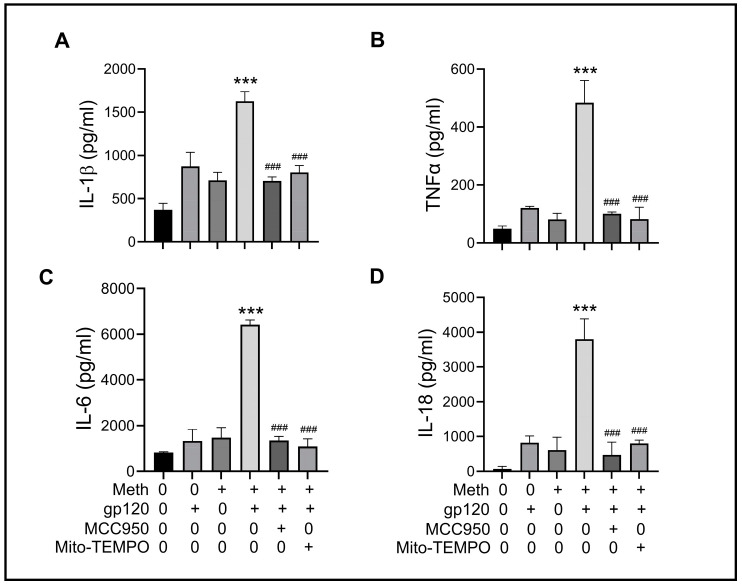
Meth increased proinflammatory cytokine production in gp120-primed microglia. The ELISA analysis results of the levels of proinflammatory cytokines in the culture supernatants recovered from microglia subjected to experimental treatments as indicated. Increased production levels of IL-1β (**A**), TNF-α (**B**), IL-6 (**C**), and IL-18 (**D**) were detected in gp120-primed, Meth-treated microglia. The pretreatment of microglia with the NLRP3 inhibitor MCC950 or the mitochondrial superoxide scavenger Mito-TEMPO significantly decreased the levels of cytokine production. *** *p* < 0.001, and ^###^
*p* < 0.001.

**Figure 6 ijms-25-03588-f006:**
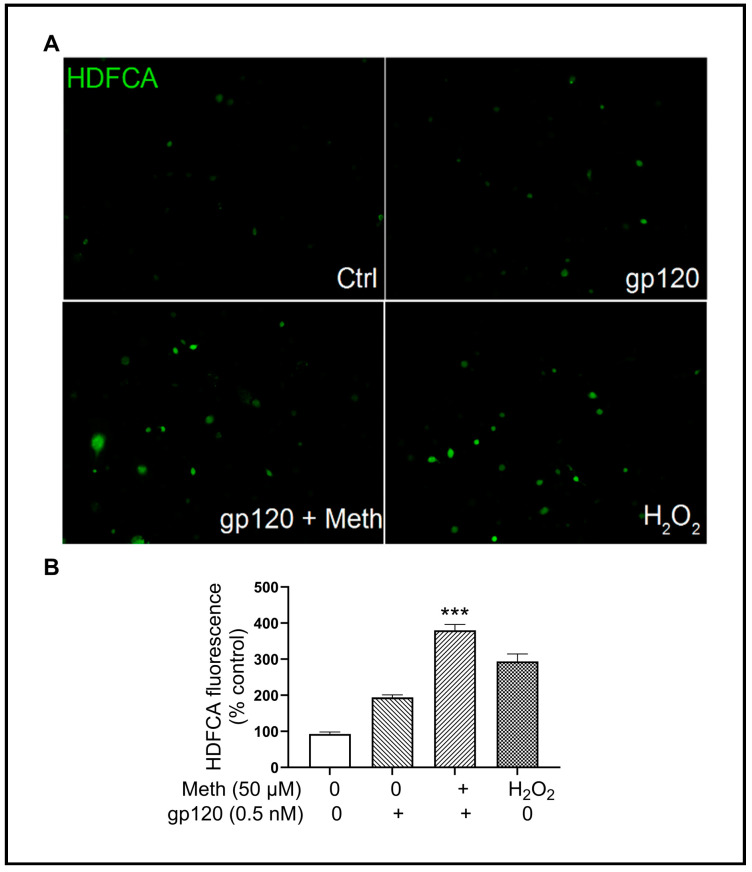
Meth and HIV-1 gp120 synergistically magnify total ROS production. Panel (**A**): Combined stimulation of Meth and gp120 synergistically activated total ROS production. Panel (**B**): Quantified results are displayed in a bar graph. All images were captured at 40× original magnification. *** *p* < 0.001.

**Figure 7 ijms-25-03588-f007:**
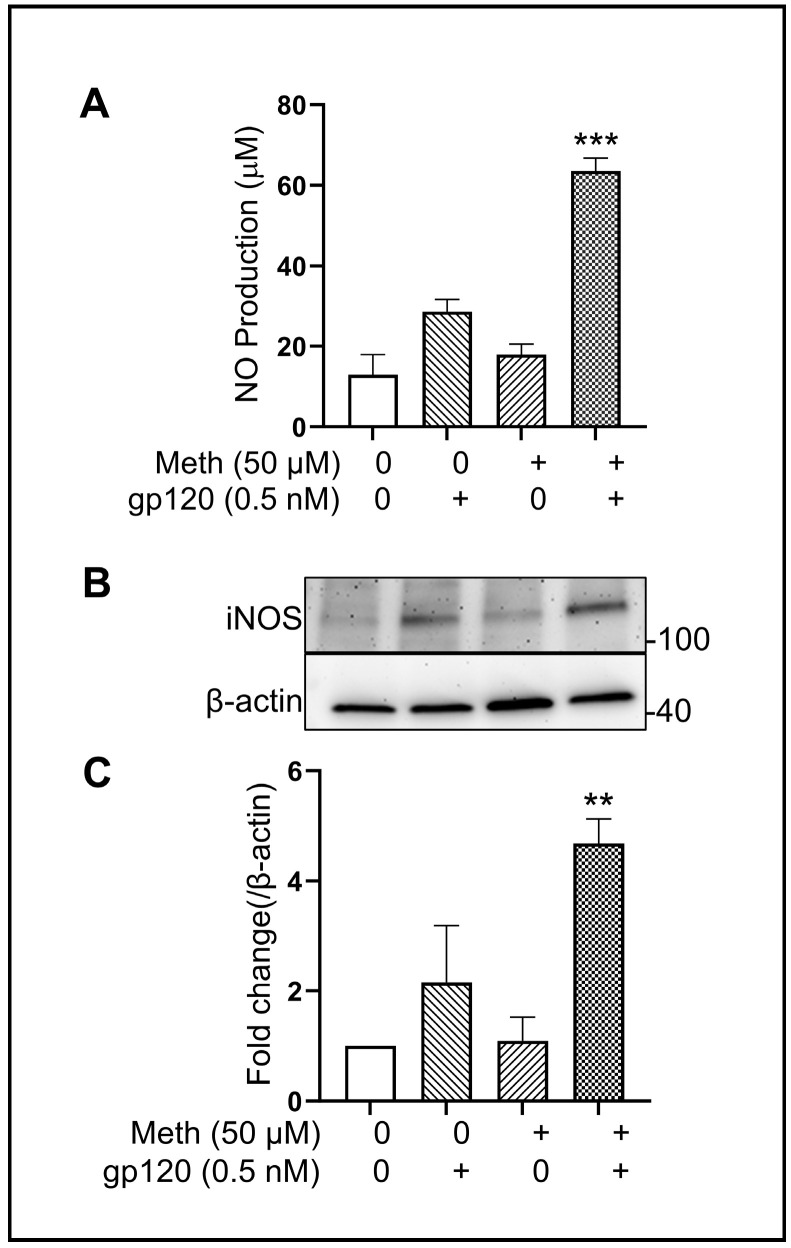
Meth-induced increased iNOS expression and NO production in gp120-primed microglia. Panel (**A**) shows Meth-induced increase in NO production in culture supernatants recovered from gp120-primed microglia. Panels (**B**,**C**) exhibits increase in iNOS expression detected in microglial lysate ** *p* < 0.01 and *** *p* < 0.001 vs. gp120-treated only control.

**Figure 8 ijms-25-03588-f008:**
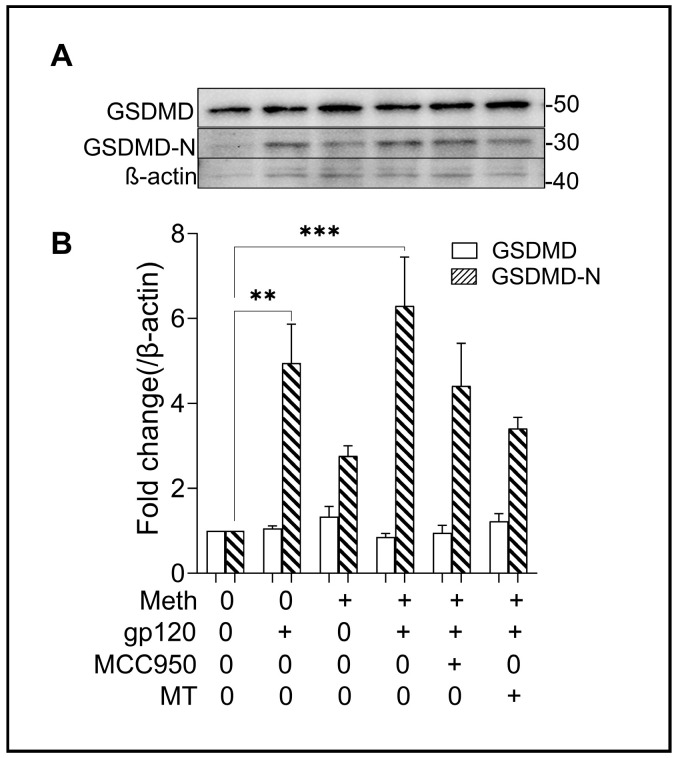
Meth-induced microglial pyroptosis. Panel (**A**) is a representative Western blot result showing that the microglial cells underwent pyroptosis after treatment with gp120, in accordance with increased N-terminal GSDMD (GSDMD-N) levels, which was further augmented by Meth treatment. The bar graph in Panel (**B**) shows the average expression levels of GSDMD and GSDMD-N. Note the augmentation effect of Meth on gp120-mediated pyroptosis. The data in Panel B represent three independent experiments, where each was performed by using technical triplicates. ** *p* < 0.01 and *** *p* < 0.001 vs. untreated control.

**Figure 9 ijms-25-03588-f009:**
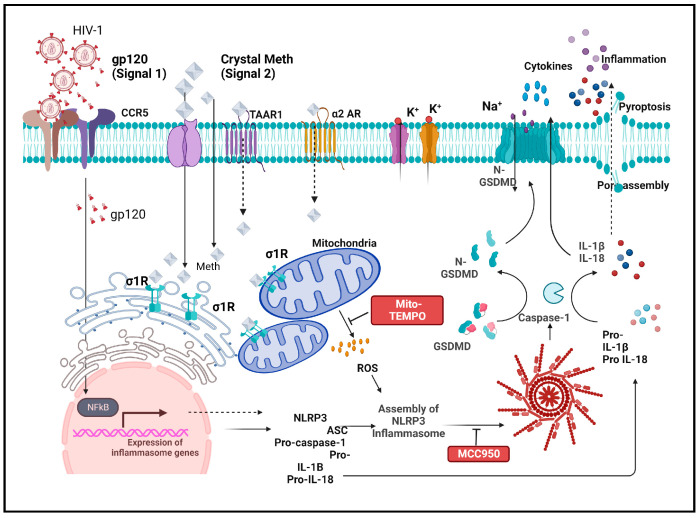
Potential mechanisms for Meth-induced enhancement of NLRP3 inflammasome activation in gp120-primed microglia in a two-signal model. Gp120 works as priming (first signal) and induces upregulation of transcription of pro-IL-1β and NLRP3. Meth helps in processing (second signal) for NLRP3 inflammasome activation via mitochondria-associated ROS signaling.

## Data Availability

The data supporting this study’s findings are available from the corresponding author upon reasonable request.

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
