# Peer review of "Methamphetamine Enhancement of HIV-1 gp120-Mediated NLRP3 Inflammasome Activation and Resultant Proinflammatory Responses in Rat Microglial Cultures"

_ijms, 2024, doi:10.3390/ijms25073588_

Round 1

Reviewer 1 Report

Comments and Suggestions for Authors

The major concern is the I-Thenticate report uploaded to the portal. It must be double checked before accepting the manuscript.

All the images must be replaced with high resolution ones.

Author may include the Cat# number of antibodies as well for ref#

I have a few comments on the Blots as well (Suppl)

1.     Few blots do not have protein markers, nor the molecular weights are seen in cropped and supp blots as well!

2.     Beta actin (Supplementary blot for Figure 8. Β‐actin) may be replaced.

Comments on the Quality of English Language

Needs editing and English proofreading

Author Response

Point 1. “The major concern is the I-Thenticate report uploaded to the portal. It must be double checked before accepting the manuscript.”

Response: We have not yet to find the I-Thenticate report uploaded to the portal, please provide us with the report for any changes needed. Thanks.

Point 2. “All the images must be replaced with high resolution ones.”

Response: Thank you for pointing this out. We prepared all figures in a high resolution (300 dpi) in the 1st submission.  According to the author guidelines all figures must be inserted into the main text close to their first citation. I am afraid that the insertion may compromise the quality of the figures. In revised submission all the images have been replaced with higher-resolution (600 dpi) versions and uploaded to the portal.

Point 3.  “Author may include the Cat# number of antibodies as well for ref#”

Response: We have included cat# for all the antibodies and reagents used in the study, and it has been updated and highlighted in the revised manuscript.

Point 4. “Few blots do not have protein markers, nor the molecular weights are seen in cropped and supp blots as well!”

Response: Thank you. All blots are now provided with molecular weights in Kd.

Point 5. “Beta actin (Supplementary blot for Figure 8. Β‐actin) may be replaced.”

Response: The image of the blot in the supplementary blot for Figure 8. Β‐actin is now improved for better visibility.

Reviewer 2 Report

Comments and Suggestions for Authors

The main problem with the contribution by Dutta et al. is the low quality of the microscopy images on which most of the conclusions rely.

Fig. 2. I simply cannot see a thing in the top row (Iba-1), nor can I detect any clear shapes in the middle row (DAPI-stained nuclei). The bottom right image (merge, gp120 + Meth) is featureless to my (otherwise normal) eye. I took my computer to a dark room, to improve perception, to no avail. These images cannot be published.

Legend to Fig. 3 reads: "Note that Meth induced co-localization of NLRP3 and caspase-1 as shown in merged images. " I'm affraid I cannot see any substantial  difference between the gb120 and the gp120 + meth images. And I don't believe in processed data, such as these images, that a machine of any kind can detect, while the eye cannot.

Similar comments can be made of the microscopy images shown in Figs. 4 and 6. Most of them are simply featureless. How can the authors say, in legend to Fig. 4, "Panel A: Representative fluorescence microscopy images display NLRP3 puncta (red) labeled by anti-NLRP3 antibody" when no red dots are even remotely visible? I am not doubting, not even for an instant, about the authors' integrity, I am complaining about the quality of the experimental data.

In general all the figures are shown at a very low resolution, I imagine this would be improved in the final, publishable, form.

Also, the statistics used are not clear to me. The authors claim "All experiments were performed in triplicate" but, what does "experiment" mean here? How many fields, of how many preparations were used to quantify the results shown, e.g. in Fig. 2C, or 4B, or 4D? This should be specified in each case.

Author Response

Point 1.”The main problem with the contribution by Dutta et al. is the low quality of the microscopy images on which most of the conclusions rely.”

Response: Thank you for pointing this out. We prepared all figures in a high resolution (300 dpi) in the 1st submission.  According to the author guidelines all figures must be inserted into the main text close to their first citation. I am afraid that the insertion may compromise the quality of the figures. In revised submission all the images have been replaced with higher-resolution (600 dpi) versions and uploaded to the portal.

Point 2. “Fig. 2. I simply cannot see a thing in the top row (Iba-1), nor can I detect any clear shapes in the middle row (DAPI-stained nuclei). The bottom right image (merge, gp120 + Meth) is featureless to my (otherwise normal) eye. I took my computer to a dark room, to improve perception, to no avail. These images cannot be published.”

Response: The images in Figure 2 exhibited visible difference in Adobe Photoshop under different experimental conditions. The discrepancy could be introduced by the insertion of the figure into the text. In revised submission, this figure has been replaced with a higher resolution (600 dpi) images in TIFF format for better visibility.

Point 3. “Legend to Fig. 3 reads: "Note that Meth induced co-localization of NLRP3 and caspase-1 as shown in merged images. " I'm affraid I cannot see any substantial difference between the gb120 and the gp120 + meth images. And I don't believe in processed data, such as these images, that a machine of any kind can detect, while the eye cannot.”

Response: Please see the prominent red (NLRP3) and green (caspase-1) colors in the merged image, which is brighter and noticeable in gp120 and meth-treated microscopy image, depicting colocalization and an indication of NLRP3 inflammasome activation. Now, we have uploaded the a higher-resolution image for better visibility.

Point 4. “Similar comments can be made of the microscopy images shown in Figs. 4 and 6. Most of them are simply featureless. How can the authors say, in legend to Fig. 4, "Panel A: Representative fluorescence microscopy images display NLRP3 puncta (red) labeled by anti-NLRP3 antibody" when no red dots are even remotely visible? I am not doubting, not even for an instant, about the authors' integrity, I am complaining about the quality of the experimental data.”

Response: Thank you for your comments.  We have increased the resolution to 600dpi for this figure and it looks much better now.

Point 5. “In general, all the figures are shown at a very low resolution, I imagine this would be improved in the final, publishable, form.”

Response: All the images have been replaced with high-resolution (600dpi) versions and uploaded to the portal.

Point 6. “Also, the statistics used are not clear to me. The authors claim "All experiments were performed in triplicate" but, what does "experiment" mean here? How many fields, of how many preparations were used to quantify the results shown, e.g. in Fig. 2C, or 4B, or 4D? This should be specified in each case.”

Response: Statistical analyses were performed by one-way ANOVA followed by post hoc Tukey’s multiple comparisons test (GraphPad Prism, version 9.4.1). A minimum p-value of 0.05 was chosen as the significance level for all tests.

In the fluorescent staining for total ROS production, nine fields under the fluorescent microscope were taken, and all cellular fluorescent intensities were quantified by Image J. The intensities of all individual cells in each field were averaged and transformed to fold changes against a control group. All experiments were performed in technical triplicate and biological triplicates unless otherwise specified.

These specifications are now included in figure legends of Fig. 2C, or 4B, or 4D

Round 2

Reviewer 2 Report

Comments and Suggestions for Authors

The manuscript has been improved accordingto my suggestions, except that, in Fig. 4, the presumed red puncta are not visible at all. The figure must be clearly improved if the paper is to be published.

Author Response

In this R2 resubmission, we have particularly revised the Figure 4 in response to your critiques. This figure illustrates methamphetamine activation of NLRP3 inflammasome in HIV-1 gp120-primed microglia as demonstrated by the presence of NLRP3 puncta visualized under fluorescence microscopy images (in red).  In previous submission, Figure 4A was assembled with 18 images in 3 columns and 6 rows. As a result, the size of each fluorescence microscopy image was not big enough, making the NLRP3 puncta not being able to be seen by naked eyes. To enhance its visibility, avoid the crowdedness and reduce its overall size of Figure 4A, we exhibited each image in a larger size and marked some (not all) of NLRP3 puncta with arrows. Thus, the new Figure 4A is composed of 3 columns and 4 rows. Nevertheless, the NLRP3 puncta can clearly be seen now in the revised figure. The two rows at the bottom were dropped off but their average results were shown in the bar graph in Figure 4B. We hope the revision on Figure 4A addresses your critiques and concerns.  Thank you very much your important critiques, which lead to an apparent improvement of the quality of this MS.

Round 3

Reviewer 2 Report

Comments and Suggestions for Authors

The revised manuscript can be recommended for publication.

Author Response

We sincerely thank you for the recommendation for publication of our manuscript.